# Utility of a Multimodal Biomarker Panel and Serum Proapoptotic Activity to Refine Diagnosis of Ovarian Adnexal Masses

**DOI:** 10.3390/diseases13100342

**Published:** 2025-10-16

**Authors:** Andrea Molina-Pineda, Francisco Osiel Jauregui-Salazar, Aleyda Guadalupe Zamudio-Martínez, Sayma Vizcarra-Ramos, Jesús García-Gómez, Benjamín González-Amézquita, Lizeth Montserrat Aguilar-Vazquez, Raquel Villegas-Pacheco, Rodolfo Hernandez-Gutierrez, Luis Felipe Jave-Suárez, Adriana Aguilar-Lemarroy

**Affiliations:** 1Centro de Investigación y Asistencia en Tecnología y Diseño del Estado de Jalisco A.C. (CIATEJ), Guadalajara 44270, Mexico; andymopi@gmail.com (A.M.-P.); rhgutierrez@ciatej.mx (R.H.-G.); 2Unidad Médica de Alta Especialidad, Hospital de Ginecología y Obstetricia, Centro Médico Nacional de Occidente, Instituto Mexicano del Seguro Social (IMSS), Guadalajara 44349, Mexico; osiel.jauregui@gmail.com (F.O.J.-S.); benggy_09@hotmail.com (B.G.-A.); raquel.villegaspa@imss.gob.mx (R.V.-P.); 3Facultad de Biología, Universidad Autónoma de Sinaloa, Culiacán 80040, Mexico; ale_zama@outlook.es; 4Centro de Investigación Biomédica de Occidente (CIBO), División de Inmunología, Instituto Mexicano del Seguro Social (IMSS), Guadalajara 44340, Mexico; sayma.vizcarra8588@alumnos.udg.mx (S.V.-R.); jesus.garcia9891@alumnos.udg.mx (J.G.-G.); lizeth.aguilar@alumnos.udg.mx (L.M.A.-V.)

**Keywords:** ovarian cancer, adnexal masses, apoptosis, HE4, CA125, sFas, sCD95, MRP8/14, OPN, SAA

## Abstract

Background/Objectives: Ovarian adnexal masses present diagnostic challenges due to their heterogeneous etiologies. Accurately differentiating these conditions is critical for timely and effective clinical intervention. This study evaluated circulating molecules and serum-induced apoptosis as complementary tools to conventional diagnostic methods (CA125, HE4, and the ROMA index) for distinguishing benign masses from malignant masses. Methods: A cohort of 136 participants (9 healthy controls, 87 women with benign ovarian adnexal masses and 40 with malignant ovarian adnexal masses) was analyzed. The induction of apoptosis in Jurkat cells by patient serum was assessed using flow cytometry. Serum concentrations of sFas/CD95, HE4, CA125, and additional molecules were measured by ELISA and LEGENDplex™. Clinical, ultrasonographic, and histopathological data were correlated with tumor malignancy. To improve diagnostic performance beyond individual biomarkers, we developed two multiparametric classifiers that integrate the dominant parameters identified through group divergence analysis and ROC evaluation across multiple clinical comparisons. Results: Malignant tumors were associated with older age (51.45 ± 8.35 years, *p* = 0.0002), postmenopausal status (61.1%, *p* = 0.0013), and larger tumor size (>10 cm). Ultrasonographic features of complexity were observed exclusively in malignant masses. Functional assays revealed reduced apoptosis in Jurkat cells exposed to malignant sera, suggesting tumor-mediated immune evasion. Although higher sFas levels were observed in tumors, no significant differences were identified between the groups. Among the circulating biomarkers, CA125, HE4, MRP8/14, OPN, and SAA levels were significantly higher in malignant tumors than in benign tumors and controls. Conclusions: The evaluation of CA125, HE4, MRP8/14, and apoptosis (Classifier 1) and, more prominently, the measurement of additional molecules: OPN, SAA, IL-6, IL-8, and IGFBP-4 (Classifier 2), systematically outperformed the ROMA. Both achieved superior specificity and balanced accuracy (Youden’s J index) across all clinical comparisons by capturing the biological diversity of malignancies.

## 1. Introduction

Ovarian cancer (OC) has been a global concern for years. In 2022, an estimated 324,603 new cases were diagnosed, and 206,956 deaths were reported, which places OC as the sixth cause of death in women worldwide [1]. Approximately 60% of women diagnosed with OC are in advanced stages, with a 5-year survival rate of 29%. Detection in early stages improves the survival rate to 92%; however, only 15% of cases are diagnosed at this stage [2,3].

The study of biomarkers has become a pivotal tool in recent years for the diagnosis and advanced clinical care of OC [4]. Among these biomarkers, the CA125 antigen, also known as mucin 16 (MUC16), is widely used for detecting epithelial ovarian cancer. Elevated CA125 levels are observed in 80% of patients with epithelial ovarian cancer but only in 50% of patients with stage I disease [5,6]. CA125 measurement is a key component for the evaluation of ovarian tumors, as levels below 20 U/mL are typically associated with benign conditions. However, levels as high as 45 U/mL may occur in non-malignant conditions such as pelvic inflammatory disease or pelvic abscesses. The established cutoff is 65 U/mL for premenopausal women and 35 U/mL for postmenopausal women. Notably, CA125 levels of 100 U/mL in premenopausal women are associated with a 21.1% probability of malignancy, whereas the same level in postmenopausal patients indicates a 74.3% likelihood. Despite its utility, CA125 has limitations: its overall sensitivity is 78.7% (range: 72–82%), and its specificity is 77.9% (range: 73.2–82.0%) [7]. Thus, it should not be used as a standalone diagnostic tool because various factors, such as pregnancy [8] and other pathologies, including gastrointestinal diseases, tuberculosis, and certain cancers (e.g., endometrial, pancreatic, and lung cancers), can also influence CA125 levels [9,10].

On the other hand, human epididymal protein 4 (HE4) is another widely used biomarker that is reported to provide better accuracy in differentiating malignant masses from benign ovarian adnexal masses, with a sensitivity and specificity of 90.5% and 87.9%, respectively [11], with significant effectiveness in advanced stages and conditions such as endometriosis. Additionally, the ROMA (Risk of Ovarian Malignancy Algorithm) [12] incorporates CA125 (HE4) and menopausal status, achieving a sensitivity of 89–92% and a specificity of 75% [13]. ROMA is considered the gold standard for risk stratification in patients with ovarian cancer [14,15,16].

One of the main pathways used by immune cells to induce apoptosis is the Fas/CD95/Apo-1 signaling pathway [17]. CD95 belongs to a subgroup of the tumor necrosis factor receptor (TNF-R) family that contains an intracellular death domain and is activated upon binding to its ligand, CD95L [17,18,19], allowing the recruitment of FADD and subsequently procaspase-8. Procaspase-8 undergoes catalytic autoactivation, initiating a caspase cascade that ultimately triggers apoptosis [19,20]. Insufficient expression of CD95 or CD95L or dysfunction of the CD95-mediated apoptotic pathway significantly contributes to the progression of various diseases, including infections, allergies, cancer, and immune disorders [21]. CD95L is present in two forms: membrane-bound (mCD95L) and soluble (sCD95L). The former induces apoptosis, and the latter induces inflammation and migration [19]. By using a cell assay system, we previously reported the presence of mCD95L in the serum of healthy individuals, indicating that blood serum normally maintains a proapoptotic microenvironment. In OC, elevated serum CD95L levels have been observed, which correlates with increased infiltration of CD3+ T cells. The latter could indicate a probable change in the proapoptotic microenvironment of serum toward inflammatory and pro-migratory conditions [22]. Moreover, a broader spectrum of circulating molecules has been implicated in shaping the ovarian tumor microenvironment. Inflammatory cytokines and chemokines, including IL-6, IL-8, CXCL9, CXCL10, and G-CSF, modulate tumor growth, angiogenesis, immune cell recruitment, and metastatic spread [23,24,25].

Adhesion molecules such as ICAM-1 and VCAM-1, along with matrix metalloproteinases (MMP-2, MMP-9), contribute to tumor invasion and extracellular matrix remodeling [26,27]. Stress- and tissue damage–associated proteins (myoglobin, NGAL, MPO, MRP8/14, and SAA) reflect processes of oxidative stress and systemic inflammation that are correlated with poor prognosis in patients with cancer [27,28,29,30,31]. In addition, osteopontin (OPN) and IGFBP-4 have been linked to chemoresistance, survival prediction, and tumor progression [32,33], whereas cystatin C has gained attention for its role in regulating protease activity and promoting metastasis [34].

The selection of these 24 molecules was therefore hypothesis-driven, aiming to integrate mediators of apoptosis, inflammation, angiogenesis, invasion, and tissue damage into a multimodal biomarker panel. By correlating their serum levels with functional apoptotic assays, our study aimed to provide a more comprehensive understanding of the circulating molecules present in patients with ovarian tumors and their potential utility in distinguishing benign masses from malignant adnexal masses.

## 2. Materials and Methods

### 2.1. Sample Collection

In this study, peripheral blood samples were collected from women with ovarian tumors who, at the time of sample collection, had not yet received a definitive diagnosis of benign or malignant adnexal masses. These women, scheduled for exploratory laparotomy in the Department of Surgical Oncology at UMAE-Hospital de Gineco-Obstetricia, Centro Médico Nacional de Occidente (CMNO, IMSS), provided blood samples after providing written informed consent. The samples were collected during the preoperative testing phase between April 2023 and December 2024. Tissue biopsies obtained during surgery were sent for histopathological analysis to determine whether the tumors were benign or malignant. All participants provided written informed consent before sample collection, and their data were handled confidentially. This study was approved by the Ethics and Research Committees of the Instituto Mexicano del Seguro Social (IMSS), CONBIOETICA-09-CEI-009-20160601, under the registration number R-2023-785-013.

### 2.2. Serum Samples

All the serum samples were obtained from peripheral blood collected by venipuncture. The blood was centrifuged at 2000 rpm for 30 min, aliquoted and stored at −80 °C until further use. The study group consisted of 136 participants with clinical and histopathological diagnoses. Among these patients, 87 had benign adnexal masses, 40 had malignant tumors, and 9 were healthy female volunteers.

### 2.3. Cell Culture

Jurkat cells were kindly provided by Prof. Henning Walczak, authenticated using the Multiplex Human Cell Line Authentication Test at Multiplexion GmbH (Friedrichshafen, Germany), and routinely cultured in RPMI 1640 growth medium (Cat. No. 22400089, Gibco, Waltham, MA, USA) supplemented with 10% heat-inactivated fetal bovine serum (Cat. No. 26140079, Gibco), 100 U/mL penicillin and 100 μg/mL streptomycin (Cat. No. 15140-122, Gibco). The cells were maintained at 37 °C in a humid atmosphere with 5% CO_2_.

### 2.4. Jurkat Cell Exposure to Serum and Apoptosis Determination

Approximately 2.0 × 10^5^ Jurkat cells were incubated per well in 12-well culture microplates and resuspended in 1 mL of complete RPMI 1640 medium and 200 µL of serum from patients with benign or malignant adnexal masses or healthy donors. The cells were incubated for 24 h at 37 °C in a 5% CO_2_ atmosphere. Additionally, a control serum sample with high Jurkat cell-killing potential and another with low potential were included in each experiment, ensuring that the percentages of apoptotic cells induced by these controls served as a reference for each assay. After the Jurkat cells were incubated with patient sera, they were individually transferred to 2 mL tubes and centrifuged at 1000 rpm for 10 min at room temperature. The supernatant was discarded, and the cell pellet was resuspended in 100 μL of incubation mixture (containing 0.5 µL of APC-Annexin V and 0.5 µL of 7-AAD) (APC Annexin V Apoptosis Detection Kit with 7-AAD, Cat. No. 640930, BioLegend^®^ Inc., San Diego, CA, USA). The samples were incubated at room temperature for 15 min in the dark. Subsequently, at least 20,000 events were analyzed on a CytoFLEX V4-B4-R3 flow cytometer (Beckman Coulter Inc., Brea, CA, USA) using the filters for the required fluorophores.

### 2.5. Detection of sFas and Proinflammatory Cytokines in the Serum

Serum concentrations of sFas and proinflammatory cytokines were determined using the LEGENDplex™ Custom Human Panel 1997 kit [CXCL8 (IL-8), IL-6, CCL11 (Eotaxin), CCL20 (MIP-3a), sFas, CXCL2 (GR0-β), CXCL10 (IP-10), CXCL9 (MIG), CXCL1 (GRO-α), and G-CSF], Cat. No. 900,007,431 and the LEGENDplex™ Human Vascular Inflammation Panel 1-S/P (12 plex) [including Myoglobin, MRP8/14, NGAL, MMP-2, OPN, MPO, SAA, IGFBP-4, ICAM-1, VCAM-1, MMP-9, and Cystatin C], Cat. No. 740590; both kits were obtained from BioLegend (San Diego, CA, USA). The capture beads from the kits were mixed and incubated with patient serum samples according to the manufacturer’s instructions. Since the beads are differentiated by size and internal fluorescence intensity, the concentrations of the molecules were assessed using flow cytometry (CytoFLEX V4-B4-R3 flow cytometer, Beckman Coulter Inc., Brea, CA, USA). The concentration of each analyte was determined using a standard curve generated by the same assay. The data were analyzed using the LEGENDplex™ Data Analysis Software Suite, which is available online (https://legendplex.qognit.com/user/login accessed on 9 September 2024).

### 2.6. Determination of Serum CA125 and HE4 Levels

Serum concentrations of CA125 and HE4 were measured using the Quantikine ELISA Human CA125/MUC16 Immunoassay (Cat. No. DCA125, R&D Systems, Inc., Minneapolis, MN, USA) and the Quantikine ELISA Human HE4/WFDC2 Immunoassay (Cat. No. DHE400, R&D Systems, Inc., Minneapolis, MN, USA), respectively. The assays were performed according to the manufacturer’s instructions, including standards, controls, and serum samples. A colorimetric detection method was used to quantify the analytes, and the plates were read using a microplate reader (Synergy HT, Cat. No. 7091000, BioTek Instruments Inc., Winooski, VT, USA) set to 450 nm with wavelength correction at 540 nm. The concentrations were calculated by comparing the optical density of the samples to a standard curve generated during the assay.

### 2.7. Statistical Analysis

Demographic differences were assessed via swarm plots to visualize age distributions across clinical groups and menopausal statuses. The calculated biomarker concentrations were compared between groups employing non-parametric statistical methods. The Kruskal–Wallis (KW) test was used to evaluate overall differences across groups, followed by pairwise comparisons with the Mann–Whitney U test. Additionally, the effect size for the KW test was quantified using the eta-square measure (η^2^). For the receiver operating characteristic (ROC) analysis, three clinically relevant comparisons were evaluated to assess different diagnostic scenarios. For each of the top biomarkers, the area under the curve (AUC) was calculated using the trapezoidal rule. All the statistical analyses were performed using Python 3.13.3 with standard scientific libraries, including pandas and numpy for data manipulation, scipy.stats and scikit-posthocs for statistical testing, and seaborn and matplotlib for visualization. Statistical significance was set at α = 0.05 for all tests. The risk of ovarian malignancy algorithm (ROMA) was calculated using the serum levels of CA125 and HE4 and menopausal status according to the following formulas:

Premenopausal women$$PI=-12.0+2.38\times ln\left(HE4\right)+0.0626\times ln\left(CA125\right)\;$$

Postmenopausal women$$PI=-8.09+1.04\times ln\left(HE4\right)+0.732\times ln\left(CA125\right)$$

Predicted Probability (PP)PP=exp(PI)/[1+exp(PI)]×100
where HE4 is the serum level of human epididymis protein 4 (pmol/L), CA125 is the serum level of cancer antigen 125 (U/mL), PI is the predictive index, and PP is the predicted probability of ovarian malignancy [6].

For classifier development, data preprocessing was employed to preserve biological relationships while allowing for the integration of parameters with variable distribution characteristics. A linear scale was preserved for CA-125, HE4, ROMA, and apoptosis values. In contrast, a log-transformation was applied to MRP8/14, OPN, SAA, IL-6, IL-8, and IGFBP-4 to normalize their distributions. The apoptosis values were negated to align with the expected directionality, ensuring that higher values corresponded to an increased risk of malignancy. Z-score standardization was subsequently applied to equalize the contributions of all biomarkers. Crucially, the scaling parameters were fitted exclusively on training data to prevent data leakage. Logistic regression was selected as the classification algorithm. Five-fold stratified cross-validation was implemented to obtain unbiased performance metrics. This process involved iterative training on 80% of the data and testing on the remaining 20%, with the final results reported as the mean AUC +/− standard deviation across all folds. For threshold selection, Youden’s J index was employed, as it identifies the optimal operating point that equally balances sensitivity and specificity across the diagnostic spectrum.

## 3. Results

### 3.1. Clinical and Ultrasonographic Characteristics of the Study Patients

A total of 136 participants were included in the study, comprising 40 patients with malignant tumors, 87 patients with benign tumors, and 9 healthy controls. Significant differences were observed among these groups regarding age, menopausal status and clinical/ultrasonographic characteristics (Figure 1). The control group consisted predominantly of premenopausal women (66.7%; mean age of 29.8 years), with postmenopausal women representing 33.3% (mean age of 59 years). Similarly, the benign group was mostly premenopausal (73.6%; mean age 36.4 years), whereas 26.4% were postmenopausal (mean age 60.7 years). In contrast, the malignant group showed the opposite trend: 37.5% were premenopausal women (mean age 43.9 years), whereas 62.5% were postmenopausal women (mean age 56.1 years). Boxplot analysis further highlighted an age-related trend, with malignant cases occurring more frequently in older, postmenopausal women, whereas benign masses were more common in younger, premenopausal women.

The ultrasonographic features of the tumor types also differed (*p*-value = 0.0441). All malignant tumors exhibited complex ultrasonographic characteristics, whereas the benign tumor group displayed a combination of both simple (12.6%) and complex (87.4%) characteristics. These findings suggest that complexity in ultrasonographic presentation may be associated with malignancy. While tumor laterality did not differ significantly between the groups (*p*-value = 0.3322), malignant tumors had higher rates of bilateral presentation (22.5%) than did benign tumors (13.8%). Unilateral presentation was more common in benign tumors (86.2%) than in malignant tumors (77.5%). Finally, tumor size differed significantly between the benign and malignant groups (*p*-value = 0.0000). The tumor size was greater for malignant tumors (12.18 ± 5.71 cm) than for benign tumors (8.3636 ± 4.21 cm). Compared with benign tumors, a greater proportion of malignant tumors exceeded 10 cm (52.55%) (21.88%), highlighting tumor size as a distinguishing factor (Table 1).

### 3.2. Prevalence and Distribution of Benign and Malignant Ovarian Adnexal Masses

The prevalence of benign and malignant ovarian adnexal masses was determined in the study population (Figure 2). Among the benign ovarian adnexal masses (*n* = 87), endometriomas were the most common, accounting for 36.8% of cases. This was followed by serous cystadenomas (20.7%), tumor-like lesions (which included ovarian stroma, cysts, granulomatous reactions due to pelvic inflammatory disease, severe endometriosis, and tubo-ovarian abscesses) in 18.4% of the patients, mature teratomas (14.9%), and mucinous cystadenomas (5.7%). Interligamentous leiomyomas (2.3%) and fibromas (1.1%) were identified with a lower frequency. Among the malignant ovarian adnexal masses (*n* = 40), high-grade serous carcinoma was the most frequently identified type, accounting for 37.5% of the cases. This was followed by endometrioid carcinoma (17.5%), mucinous carcinoma (15.0%), clear-cell carcinoma (15.0%), and low-grade serous carcinoma (10.0%). The rare malignant entities included adult-type granulosa cell tumors and Sertoli cell tumors, each representing 2.5% of the malignant cases. These findings highlight the diversity of ovarian pathologies and underscore the importance of their diagnostic characterization.

### 3.3. Sera from Patients with Ovarian Cancer Showed Reduced Apoptosis-Inducing Capacity in Jurkat Cells

We previously reported that sera from healthy individuals induce apoptosis via the CD95 pathway [35]. To assess whether this capacity is preserved in ovarian cancer patients, we used Jurkat cells (a T-lymphocyte-derived cell line recognized for its high degree of apoptosis) as apoptotic biosensors. The cells were incubated with serum samples from patients diagnosed with benign or malignant tumors, as well as serum from healthy female volunteers. The percentage of apoptotic cells was then evaluated after 24 h of incubation. As depicted in Figure 3a, sera from healthy controls induced significantly higher apoptosis levels than did sera from patients with benign or malignant tumors. As shown in Figure 3b, participants were stratified into tertiles based on apoptotic response: low (0–14.3%), medium (14.3–33.0%), and high kill (33.0–100%). The high-kill category comprised 65.1% of the benign cases but only 16.3% of the malignant cases. Conversely, malignant samples were more prevalent in the low- and medium-kill categories. Notably, no healthy controls fell into the low-kill group, and only one was classified into the medium-kill category.

Since reduced apoptosis in cancer has been linked to elevated soluble CD95 (sCD95/sFas) levels, we measured sCD95 concentrations; however, no significant differences were detected between the groups (Figure 3c). This lack of divergence may reflect the heterogeneity of ovarian tumors, variability in disease stage, or the involvement of alternative mechanisms of apoptosis resistance beyond sCD95 modulation.

### 3.4. CA125 and HE4 Levels Correlate with the ROMA Index and the Presence of Ovarian Tumors

Serum levels of CA125 and HE4 were measured by ELISA in samples obtained from healthy controls and women with benign or malignant adnexal masses. The risk of ovarian malignancy algorithm (ROMA) index was subsequently calculated using serum concentrations of CA125 and HE4, incorporating menopausal status as a variable according to the standard ROMA formula (Figure 4). CA125 levels (Figure 4a) were significantly elevated in the benign (129.6 U/mL) and malignant (478.6 U/mL) tumor groups compared with those in the control group (18.82 U/mL). Although some patients with benign tumors also presented moderately increased values, their median levels were significantly lower than those observed in the malignant group. On the other hand, the HE4 levels (Figure 4b) followed a similar trend, with the highest concentrations observed in the malignant group (4461 pg/mL). Compared with the controls, the benign patients presented a slight increase (2937 pg/mL) (2149 pg/mL). These data indicate that a progressive increase in HE4 levels is correlated with disease severity.

The ROMA index (Figure 4c) demonstrated good discriminatory capacity, with malignant cases showing substantially higher ROMA values (median ~80%) and minimal overlap with benign and control subjects. All pairwise comparisons reached statistical significance, supporting the utility of the ROMA as a diagnostic algorithm for differentiating malignant masses from non-malignant adnexal masses.

### 3.5. Serum Biomarkers Characterize Benign and Malignant Adnexal Tumors

Cytokines play crucial roles in regulating the immune response and inflammation, and their dysregulation has been implicated in various types of cancer. To determine whether these molecules have different expression patterns between benign and malignant adnexal masses, the serum levels of 24 molecules related to their biological functions, including proinflammatory cytokines, tissue damage–associated proteins, extracellular matrix remodeling enzymes, and regulatory/inhibitory proteins, were evaluated by flow cytometry in sera from healthy controls and patients with benign or malignant tumors. Figure 5 and Figure 6 show the 10 molecules with statistically significant differences among the study groups. All statistical data, including the remaining 14 molecules (which showed no significant variation), are presented in Appendix A. As depicted in Figure 5a, the levels of the proinflammatory cytokines IL-6, IL-8, and G-CSF were significantly greater in both the benign and malignant tumor groups than in the healthy control group; however, no significant differences were observed within tumor types. IL-6 levels were lowest in controls (mean 4.54 pg/mL) and elevated in benign tumors (mean 42.01 pg/mL) and malignant tumors (mean 28.80 pg/mL). Similarly, IL-8 concentrations were minimal in controls (mean 1.98 pg/mL), increased in patients with benign tumors (mean 8.79 pg/mL), and increased in patients with malignant tumors (mean 20.17 pg/mL). G-CSF levels followed a comparable pattern: lowest in controls (mean 14.53 pg/mL), elevated in patients with benign conditions (mean 96.39 pg/mL), and elevated in patients with malignant tumors (mean 79.99 pg/mL).

The levels of the tissue damage-associated proteins serum amyloid A (SAA), myeloid-related protein 8/14 (MRP8/14), and osteopontin (OPN) also differed between the groups (Figure 5b). SAA levels were substantially higher in the malignant group (mean: 31.15 ng/mL), with values reaching 97.9 ng/mL in some patients, than in the benign tumor (mean: 15.55 ng/mL) and control (mean: 12.57 ng/mL) groups. On the other hand, MRP8/14 was significantly elevated in both malignant tumors (mean 383.70 ng/mL) and benign tumors (mean 139.96 ng/mL) compared with the control group (mean 39.89 ng/mL). OPN levels were markedly elevated in patients with malignancies (mean 2002.63 pg/mL). Notably, high values exceeding 50,000 pg/mL were observed in some malignant samples. This finding contrasted with significantly lower levels in both benign tumors and healthy controls (mean 412.12 pg/mL and 297.81 pg/mL, respectively).

The extracellular matrix remodeling enzymes matrix metalloproteinase-9 (MMP-9) and -2 (MMP-2) were also evaluated, as shown in Figure 6a. Both MMP-9 and MMP-2 levels were elevated in the healthy control group (mean: 4.55 ng/mL and 1592.12 pg/mL, respectively) compared with those in the benign (2.98 ng/mL and 1354.54 pg/mL) and malignant groups (3.81 ng/mL and 813.27 pg/mL, respectively). Finally, the levels of cystatin C and insulin-like growth factor binding protein 4 (IGFBP-4) were significantly greater in the malignant group (5.20 ng/mL and 2.21 ng/mL, respectively) than in the benign group (4.18 ng/mL and 1.51 ng/mL, respectively). (Figure 6b). However, no significant differences were found compared with the control group (4.59 ng/mL and 1.94 ng/mL, respectively).

### 3.6. Comparative Analysis of Biomarker Divergence Across Clinical Groups

We also performed a comparative evaluation of the 24 soluble molecules, the ROMA (derived from CA125 and HE4 levels), and the independent death-induction assay across control, benign, and malignant ovarian tumor samples. This analysis revealed substantial variability in biomarker expression patterns between groups, as illustrated in Figure 7, which integrates both statistical significance (Kruskal–Wallis *p*-values) and effect size (η^2^). Five biomarkers demonstrated the most pronounced intergroup differences, with high statistical significance (*p* < 0.001) and large effect sizes (η^2^ > 0.14). CA125 demonstrated the greatest difference (*p* < 0.001, η^2^ = 0.311), followed by ROMA (*p* < 0.001, η^2^ = 0.249), MRP8/14 (*p* < 0.001, η^2^ = 0.245), HE4 (*p* < 0.001, η^2^ = 0.201), and apoptosis (*p* < 0.001, η^2^ = 0.164). Several inflammation-related biomarkers, including SAA (*p* < 0.01, η^2^ = 0.134), OPN (*p* < 0.01, η^2^ = 0.132), IGFBP-4 (*p* < 0.01, η^2^ = 0.129), IL-6 (*p* < 0.001, η^2^ = 0.098), and IL-8 (*p* < 0.05, η^2^ = 0.092), exhibited moderate intergroup separation, suggesting their potential value as complementary indicators. On the other hand, MMP-2 was statistically significant, with modest effect sizes (*p* < 0.05, η^2^ = 0.064). Finally, although G-CSF was statistically significant (*p* < 0.05), a minimal effect size (η^2^ = 0.035) was observed. In contrast, a group of markers, such as myoglobin, VCAM-1, NGAL, sFas, MPO, ICAM-1, cystatin C, MMP9 and various chemokines (CXCL9, CXCL10, CCL11, CXCL1, and CXCL2), exhibited minimal effect sizes (η^2^ < 0.01) and no statistical significance, indicating the inability to differentiate among the clinical groups.

### 3.7. Diagnostic Performance of Serum Biomarkers in Ovarian Adnexal Masses Classification

To evaluate the diagnostic performance of individual potential biomarkers, receiver operating characteristic (ROC) curve analysis was performed across three clinical comparisons: malignant vs. benign (malignant tumors vs. benign tumors and other tumor-like lesions), any tumor vs. control (malignant tumors, benign tumors and other tumor-like lesions vs. control), and malignant tumors vs. others (malignant tumors vs. benign tumors, other tumor-like lesions, and controls). The area under the ROC curve (AUC) was used to assess the discriminative ability of each biomarker (Figure 8).

As depicted in Figure 8a, the ROC analysis revealed in the “malignant vs. benign” comparison that CA125 (AUC = 0.833) and the ROMA index (AUC = 0.814) are the top-performing biomarkers, demonstrating excellent discriminatory ability. These were closely followed by MRP8/14 (0.778), HE4 (0.767), IGFBP-4 (0.735), SAA (AUC = 0.713), and OPN (AUC = 0.712), all of which showing good diagnostic potential. In contrast, IL-8 (AUC = 0.685), apoptosis (AUC = 0.661), and cystatin C (AUC = 0.645) exhibited limited predictive value. In the “any tumor vs. control” analysis (Figure 8b), apoptosis emerged as the best-performing biomarker (AUC = 0.895), followed by IL-6 (AUC = 0.832) and CA125 (AUC = 0.812), all of which showing excellent sensitivity and specificity rates. Other promising markers included HE4 (AUC = 0.761), G-CSF (AUC = 0.749), MRP8/14 (AUC = 0.712), and ROMA (0.708), whereas SAA, CXCL2, and OPN displayed limited discrimination (AUC < 0.70). Finally, in the “malignant vs. others” comparison (Figure 8c), consistent with previous findings, CA125 (AUC = 0.845), the ROMA index (AUC = 0.821), again ranked among the most effective discriminators, followed closely by MRP8/14 (AUC = 0.795), HE4 (AUC = 0.777), OPN and SAA (each with an AUC of 0.726). The lowest diagnostic utility was observed for IL-8, apoptosis, IGFBP-4, and IL-6 (AUC < 7).

### 3.8. Development and Validation of Multiparametric Classifiers for Enhanced Prediction of Ovarian Adnexal Mass

To improve diagnostic performance beyond individual biomarkers, we developed two multiparametric classifiers that integrate the dominant parameters identified through group divergence analysis and ROC evaluation across multiple clinical comparisons (Figure 9). These classifiers were designed to systematically augment the established ROMA by incorporating complementary biologically relevant features. Classifier 1 consists of a compact panel of four parameters: CA-125, HE4, MRP8/14, and apoptosis. This combination retains core clinical biomarkers while incorporating key inflammatory and apoptotic regulators. Classifier 2 expands this baseline to nine features, adding OPN, SAA, IL-6, IL-8, and IGFBP-4 to the initial set, thereby including a broader spectrum of inflammatory cytokines, acute-phase proteins, and growth factors.

This feature selection strategy prioritized parameters that showed consistent performance across different clinical comparisons while ensuring coverage of diverse biological pathways. CA-125 and HE4 were included as established clinical standards, whereas apoptosis was incorporated because of its unique—although inversely correlated—biological significance, providing complementary diagnostic value. Both classifiers were evaluated against ROMA across the same three clinical scenarios used for individual biomarker assessment. ROMA performance was evaluated via conventional ROC analysis, whereas the multiparametric classifier employed cross-validation to estimate robust real-world performance.

In the “malignant vs. benign” comparison (Figure 9a), ROMA showed baseline performance (AUC = 0.814), with 77.1% sensitivity and 78.5% specificity at a cutoff of 51.52%. Classifier 1 achieved modest improvement (AUC = 0.831 +/− 0.112), with 80.0% sensitivity and 84.8% specificity, increasing the Youden’s J index from 0.556 to 0.648. Classifier 2 exhibited superior performance (AUC = 0.880 +/− 0.084), maintaining 80.0% sensitivity while markedly increasing specificity to 93.7% (Youden’s J = 0.737).

The most pronounced improvement was observed in the “any tumor vs. control” scenario (Figure 9b). ROMA showed limited utility (AUC = 0.708), with only 40% sensitivity despite 100% specificity (Youden’s J = 0.404). Classifier 1 demonstrated exceptional performance (AUC = 0.947 +/− 0.036), achieving 89.5% sensitivity and 88.9% specificity (Youden’s J = 0.784). Classifier 2 also showed enhanced performance (AUC of 0.917 ± 0.113), with 83.3% sensitivity and 100% specificity, yielding the highest balanced accuracy in this scenario (Youden’s J = 0.883).

In the “malignant vs. others” comparison (Figure 9c), ROMA again provided a baseline value (AUC = 0.821, sensitivity = 77.1%, and specificity = 80.7%; Youden’s J = 0.578). Classifier 1 showed comparable discrimination (AUC = 0.830 +/− 0.125), with slightly improved sensitivity (80.0%) and specificity (86.4%); Youden’ J = 0.664. Classifier 2 demonstrated superior performance capacity (AUC = 0.896 +/− 0.091), achieving 82.9% sensitivity and 90.9% specificity, with the highest balanced accuracy (Youden’s J = 0.738). The detailed statistical results for these comparisons can be found in Table 2.

## 4. Discussion

Ovarian adnexal masses present a significant diagnostic challenge due to their diverse etiologies, which range from benign cysts to malignant tumors. Approximately 60% of ovarian cancer patients are diagnosed at advanced stages, with an overall 5-year survival rate of only 29%. In contrast, early-stage disease has an overall 5-year survival rate of 92% [3]. Early and accurate differentiation between benign and malignant masses is critical for guiding treatment decisions and improving patient outcomes. While traditional diagnostic tools (such as imaging techniques and clinical evaluations) are invaluable, they often prove insufficient, particularly in borderline cases [7,36,37].

As visualized in Figure 1, our data revealed that age and menopausal status distributions differed markedly among the control, benign, and malignant ovarian cases. Benign lesions were more common in younger, premenopausal women (mean age: 36.4 years), whereas malignant cases predominantly affected postmenopausal women (mean age: 56.3 years), which is consistent with established epidemiological trends [2]. Advanced age and postmenopausal status are well-documented risk factors for epithelial ovarian cancer, which primarily occurs in women over 50 years of age. This association may reflect cumulative ovulatory cycles, hormonal changes, and age-related molecular alterations in the ovarian epithelium [38].

According to the histopathology results, 71% of the tumors were benign, whereas 29% were malignant (as visualized in Figure 2). These findings align with data from Turkey, which reported a similar distribution (74.9% benign vs. 25.1% malignant) [39]. In contrast, studies from Nigeria indicate a greater proportion of malignant tumors (54.9%) than benign tumors (41.2%) [40]. Regarding ovarian adnexal masses subtypes, the most common benign diagnosis in our study was endometrioma, followed by serous cystadenoma and mature teratoma. Among malignant tumors, high-grade serous carcinoma was the most common, followed by endometrioid, mucinous, and clear-cell carcinomas. Similar trends in malignant tumors have been reported in Italy [41] and Turkey [39]. However, in a Nigerian study, the proportion of malignant tumors differed significantly, with a high proportion of serous and mucinous tumors and a decreasing number of endometrioid and clear-cell carcinomas [40]. In contrast to Mexico, low rates of endometrioid carcinoma have also been reported in China and the United States [42,43,44]. The geographic variations in histologic subtypes in our malignant cohort underscores that our findings may be particularly relevant for populations with a similar pathological landscape, highlighting the need for region-specific diagnostic strategies.

The primary aim of our study was to determine the ability of patient serum to induce apoptosis by using a cell system as a biosensor (Jurkat cells), as demonstrated previously [45,46]. As shown in Figure 3, our findings revealed a significant disparity in apoptotic capacity between controls and patients’ sera. Sera from healthy volunteers efficiently induced apoptosis in Jurkat cells (64.98 ± 23.7% apoptosis). In contrast, sera from women with benign or malignant tumors showed reduced apoptotic activity, with rates ranging from 27.54% to 20.44%, respectively. Serum proapoptotic activity is controlled by the FAS pathway and modulated by sFas levels [47]. In 2008, Tamakoshi et al. [48] measured soluble FAS (sFas) levels in 2353 healthy individuals and 798 patients with different cancer types. They reported levels of 2.41 ± 1.81 ng/mL in the healthy group, whereas cancer patients presented levels ranging from 2 to 6 ng/mL, with some cases reaching concentrations as high as 9–11 ng/mL [48]. Additionally, a five-year follow-up of these patients revealed that serum sFas levels are not only a prognostic factor in cancer but also a biomarker capable of identifying individuals at high risk of developing this pathology. Low proapoptotic activity or high sFas levels may indicate crucial immune evasion mechanisms, potentially enabling tumors to circumvent antitumor immune responses.

In addition, a study by Hefler L. et al. analyzed serum sFas levels in healthy patients (mean level of 1.5 ng/mL), patients with benign ovarian tumors (mean level of 2.3 ng/mL), and patients with malignant ovarian tumors (mean level of 3.7 ng/mL). The study concluded that elevated sFas combined with CA125 levels (mean 16.6 U/mL in benign tumors vs. 343 U/mL in malignant tumors) could aid in distinguishing benign ovarian cysts from ovarian cancer [49]. Similarly, Ma Yaxi et al. analyzed sFas levels and T-cell apoptosis in peripheral blood and peritoneal fluid. They reported that sFas levels were significantly higher in stage III-IV ovarian carcinoma than in stage I-II ovarian carcinoma and benign ovarian tumors [50]. Elevated serum sFas levels have also been observed in patients with breast cancer [51], colon cancer [52], renal [53,54], cervical, endometrial, and ovarian carcinomas [55]. Our study did not find statistically significant differences in sFas levels among the control group, benign tumor group, and malignant tumor group. However, women with tumors tended to have slightly higher levels of this marker (Figure 3). These findings underscore the variability described in previous reports. While some studies have shown higher sFas levels in advanced ovarian cancer, others have not detected clear differences across tumor subtypes or between benign and malignant tumors. Our results are in line with the latter, suggesting that sFas may have limited value in distinguishing between ovarian tumor categories, although its role in reflecting disease stage or progression cannot be dismissed.

The incorporation of molecular biomarkers, particularly HE4 and CA125, into clinical diagnostic algorithms has markedly improved the discrimination between benign and malignant ovarian masses [9,10]. A comprehensive meta-analysis of 23 studies published between 2004 and 2021 involving 10,594 epithelial ovarian cancer cases demonstrated that CA125 levels could predict disease progression risk. High serum levels were associated with poor survival outcomes [56]. In our study, CA125 levels aligned with those previously reported by different research groups [57,58], where the average concentration in control patients was 18.82 U/mL, whereas in benign and malignant patients, the levels were 129.6 U/mL and 478.6 U/mL, respectively (Figure 4a). Furthermore, Sharma Muhammad et al. (2023) highlighted the value of combining HE4, CA125, and IL-6 for predicting tumor resectability in advanced epithelial ovarian cancer, with HE4 serving as a robust marker for tumor burden and aggressiveness [59].

On the other hand, the results of the present study confirmed that HE4 levels are significantly elevated in patients with malignant ovarian tumors compared with those with benign lesions and healthy controls (Figure 4b). These findings are consistent with the literature supporting the role of HE4 as a sensitive biomarker for ovarian malignancy. In a comprehensive multicenter study by A. Hada et al. (2020), HE4 demonstrated superior specificity over CA125 in differentiating benign from malignant pelvic masses, particularly in premenopausal women and in those with early-stage disease, where elevated HE4 levels were observed in 50–89% of patients depending on the histologic subtype [60].

These findings reinforce the growing clinical utility of HE4 in combination with CA125, as recently reported by different studies, which concluded that dual-marker strategies significantly increase diagnostic sensitivity and specificity for early-stage ovarian cancer compared with CA125 alone [61,62,63,64,65]. Based on the individual performance of CA15 and HE4, the ROMA index includes both markers and menopausal status. ROMA is considered the gold standard for risk stratification in patients with ovarian cancer. In our study, as visualized in Figure 4c, we observed a significant difference in the ROMA index between controls and patients with malignant tumors, as well as between patients with benign and malignant tumors. To further analyze these findings, we calculated the optimal ROMA cutoff values by maximizing the Youden index, optimizing the balance between sensitivity and specificity for discriminating malignant from benign adnexal masses and tumor-like lesions. The optimal cutoff value was 51.52, yielding a sensitivity of 77% and a specificity of 78%.

In agreement with our data (as shown in Figure 5a), elevated IL-6 levels in serum [66,67] and plasma [68] have been widely documented in ovarian cancer patients and are associated with poor prognosis [69,70]. Although IL-6 has potential as a biomarker for discriminating healthy individuals from ovarian cancer patients, our findings reveal its limited utility in distinguishing benign from malignant ovarian conditions, a finding also reported by Kampan et al. [66]. However, IL-6 may enhance diagnostic accuracy when combined with other serum biomarkers. For instance, its integration with CA125 has been proposed to improve the differentiation between endometriomas and malignant ovarian tumors. Combined with CA125 and HE4 augments the sensitivity, specificity, and accuracy of these tests [66]. Moreover, Han et al. proposed a biomarker panel comprising CA125, HE4, IL-6, and E-CAD that can distinguish between early-stage ovarian cancer patients and noncancer controls with a specificity of 95–100% [71].

Our study revealed significantly elevated serum levels of IL-8 in patients with malignant ovarian tumors compared with healthy controls, which is consistent with previous reports [66,69,72,73,74,75]. However, consistent with the results of Kampan et al. [66], we observed no significant difference in IL-8 levels between the benign and malignant groups. These findings suggest that while proinflammatory cytokines such as IL-8 reflect tumor-associated inflammation, they may lack sufficient specificity to differentiate between benign and malignant pelvic masses when used individually. Nevertheless, conflicting evidence exists regarding the diagnostic utility of IL-8. Pawlik et al. proposed IL-8 as a useful biomarker to distinguish between benign and malignant states because they reported increased serum levels of cytokines/chemokines in ovarian cancer patients compared with those in benign ovarian cystic patients [75]. In addition, Crispim et al. proposed that IL-8 can be incorporated into a cytokine panel with IL-6 and IL-10 to distinguish endometriomas from ovarian malignant tumors [74].

The chemokine IL-8 (CXCL8) has also been associated with poor prognosis and reduced survival in advanced epithelial ovarian cancer patients [73]. Recent studies have proposed that when combined with CA125 and IL-6, IL-8 may serve as a clinically useful biomarker panel for guiding treatment decisions and personalizing therapeutic approaches [76].

The present study revealed significantly elevated serum levels of granulocyte colony-stimulating factor (G-CSF) in patients with malignant ovarian tumors compared with healthy controls, but no significant difference was detected between the benign and malignant groups. The literature presents conflicting data regarding the role of G-CSF in ovarian cancer. Horala et al. reported elevated G-CSF levels in type II ovarian cancer patients, correlating them with increased angiogenesis markers and tumor aggressiveness [24]. Conversely, Lawicki et al. reported that ovarian cancer patients in stages I and II presented lower plasma G-CSF levels than healthy controls did [77]. Notably, Lokshin et al. demonstrated that a multimarker panel incorporating G-CSF with CA125, CA19-9, EGFR, eotaxin, IL-2R, cVCAM, and MIF achieved a high-performance biomarker panel for early-stage ovarian cancer detection, with a sensitivity of 98.2% and a specificity of 98.7% [78]. Furthermore, G-CSF expression in tumor cells and the surrounding stroma is not an adverse prognostic factor in patients with ovarian cancer [79].

We also found that SAA levels were significantly greater in patients with malignant ovarian tumors than in patients with both benign tumors and healthy controls. Our results are consistent with those of previous reports showing that SAA is overexpressed in epithelial ovarian cancer (EOC) and may contribute to tumor progression. Li et al. (2020) demonstrated that high serum SAA levels were associated with advanced FIGO stage, lymph node metastasis, and poor survival in EOC patients, indicating its potential prognostic value [31]. Similarly, SAA has been proposed as part of a serum biomarker panel that could improve diagnostic accuracy for ovarian cancer, particularly when combined with CA-125 and HE4 [80]. On the other hand, SAA has been shown to promote the proliferation, migration, and invasion of ovarian cancer cells in vitro, suggesting that SAA may contribute to the aggressive behavior of malignant cells. Lin et al. (2019) reported that SAA stimulated epithelial–mesenchymal transition (EMT) in ovarian cancer cell lines, further supporting its role in tumor progression and metastasis [81]. Moreover, elevated expression of SAA has also been detected in ovarian cancer tissue, particularly in high-grade serous carcinoma, and is associated with poor differentiation and increased inflammatory infiltration [82].

Additionally, we determined that the serum levels of MRP8/14 (calprotectin) were significantly greater in patients with malignant ovarian tumors than in those with benign tumors and healthy controls. Although the median value in the benign group was similar to that in the control group, the elevated level in the malignant group suggested that MRP8/14 may serve as a useful biomarker for distinguishing malignant from nonmalignant ovarian disease. Our findings align with previous work by Ødegaard et al., who reported increased circulating calprotectin levels in ovarian carcinomas and borderline tumors, with levels correlated with tumor type and aggressiveness [83]. However, the roles of MRP8 (S100A8) and MRP14 (S100A9) appear to be dual: they can promote tumor progression through inflammation and immune modulation but also induce cytotoxicity and apoptosis under certain conditions, as shown by Viemann et al., who reported that MRP8/14 disrupts endothelial integrity and triggers cell death [84]. Interestingly, high MRP8 expression has been associated with better survival in serous and advanced-stage ovarian cancer patients, whereas MRP14 has no prognostic impact [85].

Consistent with previously reported data, we observed that serum OPN levels were significantly higher in patients with malignant ovarian tumors than in those with benign tumors and healthy controls, with some malignant cases reaching values above 50,000 pg/mL. This wide dynamic range supports the potential of OPN as a sensitive biomarker for malignancy. Our findings align with those of Živný et al., who reported that OPN levels were significantly elevated in serous ovarian carcinoma compared with borderline tumors and that OPN outperformed CA125 in diagnostic accuracy [86]. Similarly, a meta-analysis by Hu et al. confirmed that OPN has high sensitivity (66%) and specificity (88%) for ovarian cancer diagnosis, with an AUC of 0.85, supporting its use in noninvasive detection [32]. In addition to its diagnostic value, OPN is implicated in tumor progression. Hu et al. (2019) reported that OPN gene expression was correlated with increased cell proliferation and reduced apoptosis in ovarian tumors, suggesting a direct role in tumor growth [87]. Additionally, Rani et al. reported elevated OPN levels in advanced and high-grade tumors, supporting its relevance as a prognostic indicator [88].

Our results revealed that serum MMP-2 levels were lower in patients with malignant ovarian tumors than in those with benign tumors and controls. This finding contrasts with previous reports that associated high MMP-2 expression with poor prognosis and advanced disease [89,90]. This discrepancy suggests that serum MMP-2 may not accurately reflect intratumoral expression.

In the case of MMP-9, we observed a slight increase in malignant cases versus benign cases, but the levels remained lower than those in controls. Previous studies linked elevated MMP-9 in tissue with poor outcomes [91,92]. Local MMP activity in the tumor microenvironment is critical, and circulating levels may not capture this dynamic, limiting their standalone diagnostic value [93].

On the other hand, serum cystatin C levels are elevated in patients with malignant ovarian tumors compared with those with benign tumors and healthy controls. Different studies have shown that the balance between cysteine proteases (such as cathepsin B) and their inhibitor cystatin C is critical in regulating ovarian cancer invasion. While cathepsin B promotes extracellular matrix degradation, cystatin C may act as a compensatory response aimed at limiting tumor progression [34,94]. Thus, the elevated cystatin C levels observed in our malignant group may reflect this protease-inhibitor imbalance rather than being directly tumor suppressive.

Finally, in our analysis, serum IGFBP-4 levels were higher in patients with malignant ovarian tumors than in those with benign tumors and controls. This finding aligns with previous findings that reported that IGFBP-4 is consistently elevated in tumor tissue and serum across all stages of epithelial ovarian cancer [95]. Additionally, IGFBP-4 has been proposed as a prognostic marker [96], while its expression has been associated with endocrine responsiveness [97], highlighting its clinical relevance beyond diagnosis.

Importantly, these findings suggest that a multimodal approach—combining functional assays such as apoptosis with established molecular biomarkers and imaging features—could enhance clinical decision-making. In line with this, the multiparametric classifiers developed here were designed to augment the performance of the established ROMA by incorporating complementary biologically relevant parameters. Classifier 1, which integrates CA-125, HE4, MRP8/14, and apoptosis, achieved consistent improvements across all the clinical scenarios. This finding indicates that a compact panel can provide superior discriminatory power compared with ROMA alone. Classifier 2, which expands the panel to nine biomarkers, demonstrated the most pronounced improvements. This was particularly evident in differentiating tumor samples from healthy controls, where both the sensitivity and specificity were markedly enhanced. These improvements translated into significant increases in Youden’s J index, underscoring the potential of multiparametric strategies to achieve balanced diagnostic accuracy.

From a biological perspective, the inclusion of inflammatory cytokines (IL-6, IL-8), acute-phase proteins (SAA), and growth regulators (IGFBP-4, OPN) reflects the multifactorial nature of ovarian tumorigenesis. This process involves a complex interplay of immune dysregulation, tissue remodeling, and apoptosis. By considering this biological diversity, Classifier 2 achieved superior discriminatory performance across benign, malignant, and control comparisons. Notably, these improvements were achieved without compromising sensitivity, suggesting that multiparametric classifiers could reduce false positives and, consequently, the risk of unnecessary surgical interventions.

This study has some limitations. First, the use of whole patient serum for cell incubation restricts the ability to identify the specific bioactive molecules responsible for the observed apoptotic effects. Future studies should employ purified or recombinant components (e.g., individual cytokines or sFas) to clarify their specific contributions. Second, while we quantified the molecule levels using the LEGENDplex™ platform, few recent studies have used this methodology, limiting the comparability of our results with the current literature. Validation using more widely established, highly sensitive techniques would strengthen the robustness and translational relevance of the findings.

## 5. Conclusions

This study highlights the utility of combining clinical, imaging, and biomarker data for ovarian cancer diagnosis. In terms of clinical and ultrasonographic features, malignant tumors were more prevalent in older, postmenopausal women, whereas benign masses predominated in younger, premenopausal women. In addition, malignant tumors are larger in size (>10 cm) and have complex ultrasonographic features, whereas benign tumors have both simple and complex morphologies.

In terms of histological distribution, endometriomas (36.8%) and serous cystadenomas (20.7%) were the most common benign tumors, whereas high-grade serous carcinoma (37.5%) was the predominant malignancy.

Notably, compared with those from benign and control samples, sera from malignant samples induced significantly lower levels of apoptosis in Jurkat cells. ROC analysis revealed that apoptosis exhibited superior discriminatory power (AUC = 0.895) in differentiating “any ovarian tumor from controls”, highlighting its potential as a broad screening tool.

With respect to the diagnostic biomarkers, we detected significantly higher serum levels of CA125, HE4, IL-6, IL-8, G-CSF, SAA, MRP8/14, OPN, MMP9, and MMP2 in women with benign or malignant tumors than in healthy controls. However, CA125 had the greatest discriminatory power in differentiating “malignant vs. benign” or “malignant vs. others”, followed by ROMA, MRP8/14, HE4, OPN, and SAA.

The combined evaluation of CA125, HE4, MRP8/14, and apoptosis, along with the measurement of OPN, SAA, IL-6, IL-8, and IGFBP-4, systematically outperformed the ROMA in all clinical comparisons. These findings support a multiparameter approach to improve the diagnostic accuracy of ovarian adnexal mass evaluation.

## Figures and Tables

**Figure 1 diseases-13-00342-f001:**
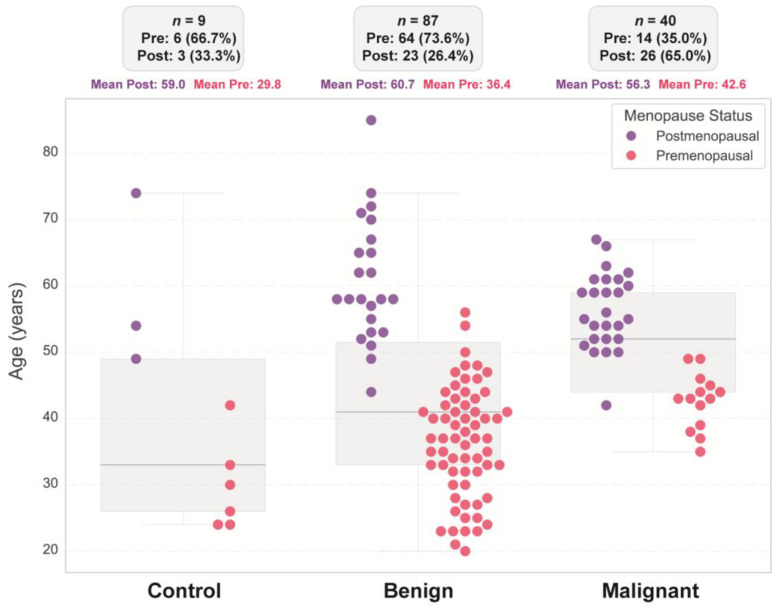
Age distribution stratified by diagnostic group and menopausal status. Individual data points represent age values for each group (control, benign, and malignant), color-coded by menopausal status (premenopausal: pink; postmenopausal: purple). Box-and-whisker plots display the median (center line) and the interquartile range (25th–75th percentiles) for age in each group. Summary statistics above each group indicate the total number of patients (*n*), the number and percentage of pre- and postmenopausal women, and their respective mean ages.

**Figure 2 diseases-13-00342-f002:**
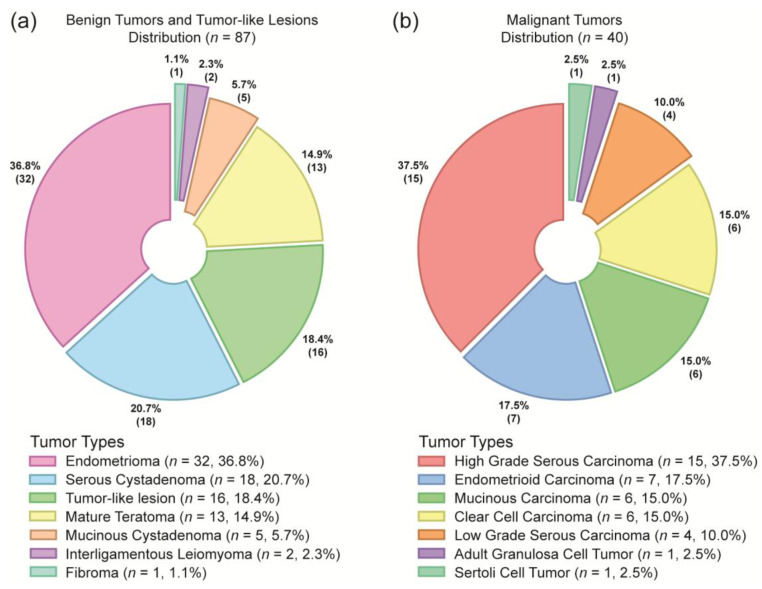
Histological distribution of ovarian tumor types in the benign and malignant groups. Pie charts illustrate the proportions of different tumor subtypes in participants diagnosed with (**a**) benign (*n* = 87) and (**b**) malignant (*n* = 40) tumors, with each category represented by distinct colors and labeled with absolute frequency and percentage.

**Figure 3 diseases-13-00342-f003:**
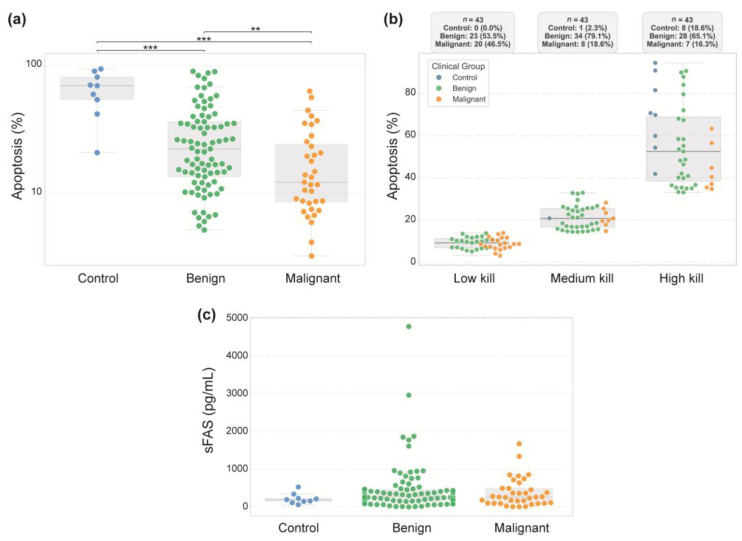
Apoptosis induction and sFas levels in patients. The percentage of apoptotic Jurkat cells was measured after 24 h of incubation with sera from healthy controls and patients with benign or malignant adnexal masses. Apoptosis was assessed by flow cytometry using 7-AAD/PI staining, with a minimum of 20,000 events analyzed per sample. (**a**) Percentage of apoptosis induction by control (*n* = 9), benign (*n* = 87), and malignant (*n* = 40) groups. Statistical analysis revealed significantly less apoptosis in the malignant group than in the benign and control groups (*** *p* < 0.001, ** *p* < 0.01). (**b**) Distribution of patients categorized into low, medium, and high apoptotic levels on the basis of tertiles of the percentage of apoptotic cells. The proportion of individuals in each clinical group is shown for each kill level. (**c**) sFas/sCD95 levels were measured by flow cytometry. sFas concentrations are expressed in pg/mL. Box-and-whisker plots display the median and the 25th–75th percentiles for controls, benign tumors, and malignant tumors.

**Figure 4 diseases-13-00342-f004:**
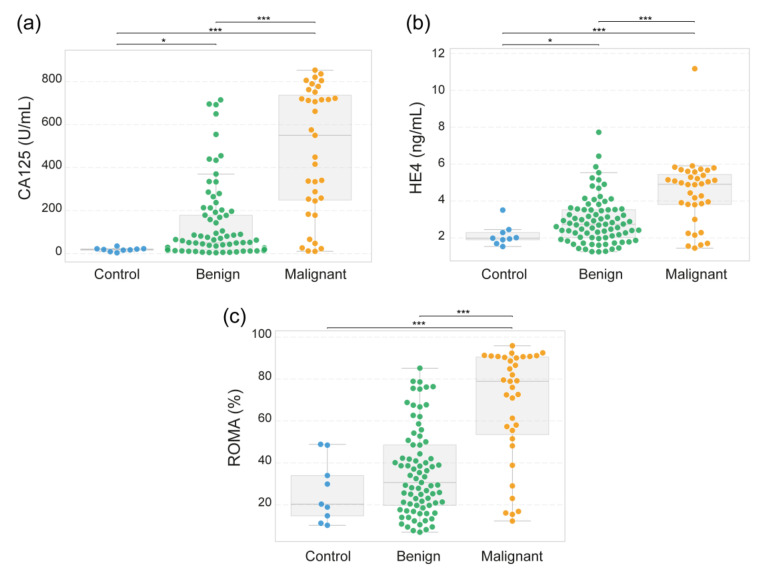
Serum levels of CA125, HE4 and ROMA index. CA125 (**a**) and HE4 (**b**) levels were measured by ELISA in sera from healthy controls and women with benign or malignant adnexal masses. (**c**) The ROMA (Risk of Ovarian Malignancy Algorithm) index was determined considering CA124 and HE4 levels. The data are represented as dot plots overlaid with box-and-whisker plots indicating the median and 25th–75th percentiles. Each dot represents an individual subject. Statistical significance is represented by * *p* < 0.05, and *** *p* < 0.001.

**Figure 5 diseases-13-00342-f005:**
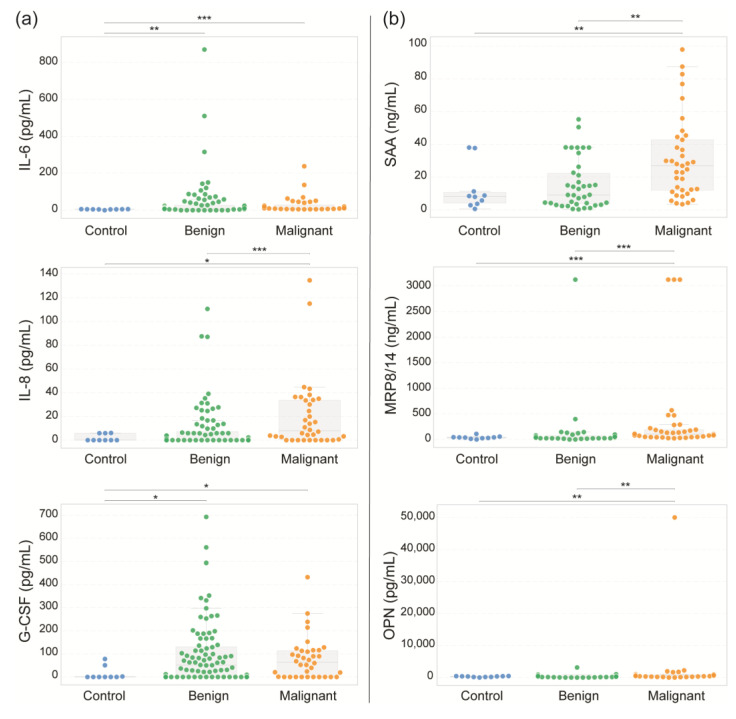
Serum concentrations of proinflammatory cytokines and damage-associated proteins. Serum concentrations of IL-6, IL-8, and G-CSF (**a**) and SAA, MRP8/14, and OPN (**b**) were determined in samples from healthy controls and women with benign or malignant adnexal masses using flow cytometry. Box-and-whisker plots display the median (central line), 25th–75th percentiles (box), and full data ranges (whiskers). Statistical significance is represented by * *p* < 0.05, ** *p* < 0.01, and *** *p* < 0.001.

**Figure 6 diseases-13-00342-f006:**
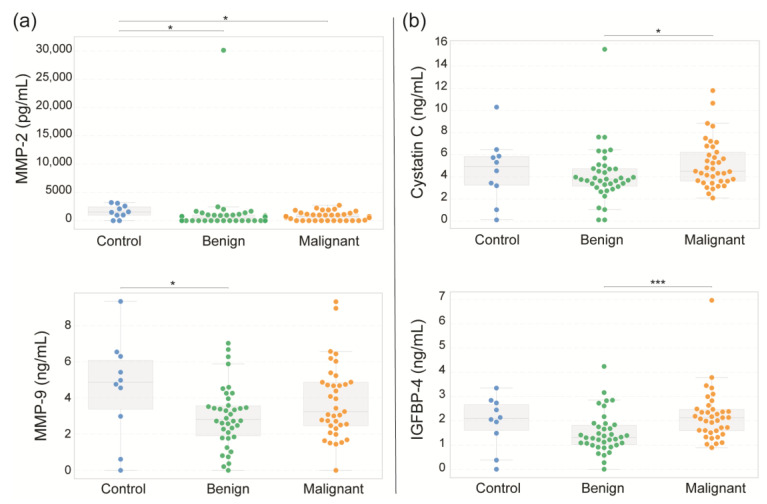
Serum concentrations of extracellular matrix remodeling enzymes and regulatory proteins. Serum concentrations of MMP-9, MMP-2, OPN (**a**), Cystatin C and IGFBP-4 (**b**) were determined in samples from healthy controls and women with benign or malignant adnexal masses using flow cytometry. Box-and-whisker plots display the median (central line), 25th–75th percentiles (box), and full data ranges (whiskers). Statistical significance is represented by * *p* < 0.05, and *** *p* < 0.001.

**Figure 7 diseases-13-00342-f007:**
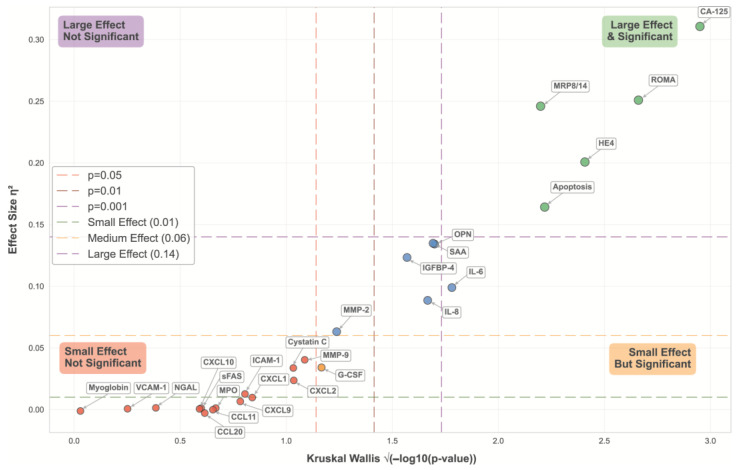
Statistical significance versus effect size analysis of serum biomarkers across ovarian clinical groups. The graph displays the relationship between statistical significance (Kruskal–Wallis *p*-values, x-axis) and effect size (η^2^, y-axis) for the 26 biomarkers analyzed in control, benign, and malignant ovarian tissue samples. Points are color-coded according to their combined significance. Red circles: small effect size with no statistical significance; yellow circles: small effect size but statistical significance; blue: medium effect size with statistical significance; and green circles: large effect size and high statistical significance.

**Figure 8 diseases-13-00342-f008:**
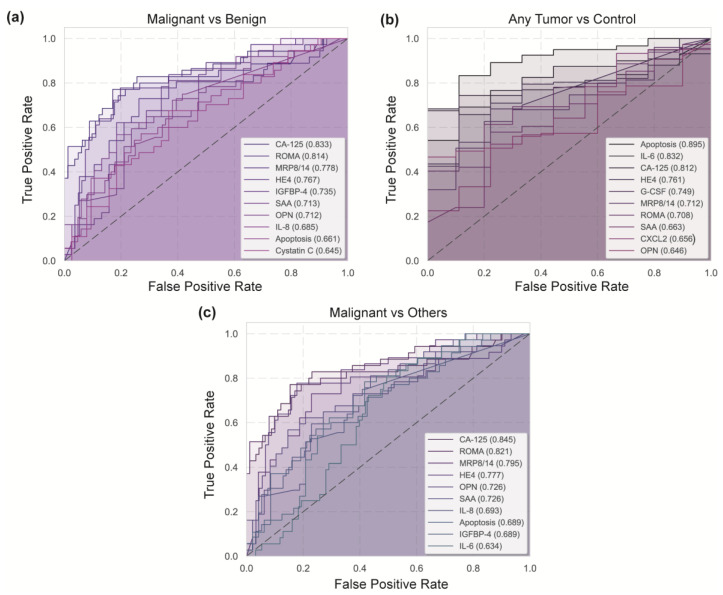
ROC curve analysis of serum biomarkers for ovarian adnexal masses discrimination. ROC curves depicting the diagnostic performance of serum biomarkers for distinguishing between groups in three comparisons: (**a**) malignant vs. benign tumors, (**b**) any tumor (benign or malignant tumor) vs. control, and (**c**) malignant vs. others (control/benign tumors/tumor-like lesions). The area under the curve (AUC) is shown for each biomarker in parentheses within the figure legends. Higher AUC values indicate better diagnostic performance.

**Figure 9 diseases-13-00342-f009:**
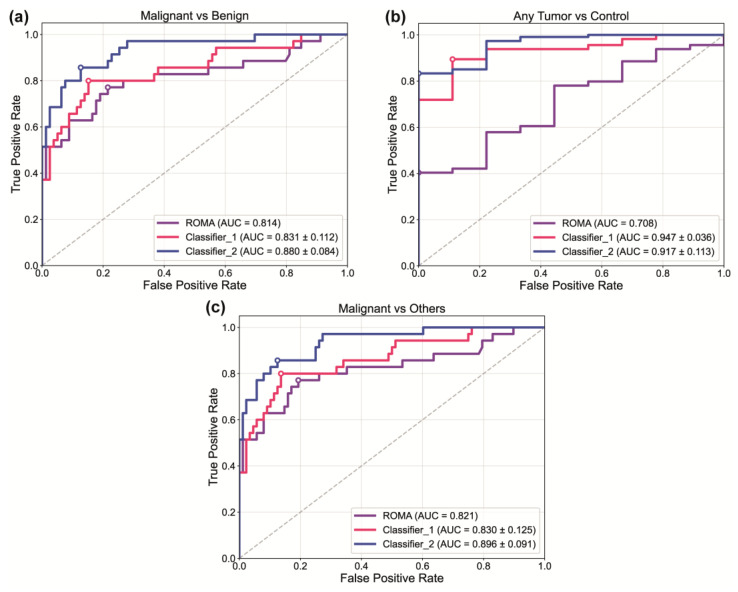
ROC curve analysis of multiparametric classifiers for ovarian adnexal masses discrimination. Receiver operating characteristic (ROC) curves comparing the diagnostic performance of the ROMA and two multiparametric classifiers across three clinical comparisons: (**a**) malignant vs. benign tumors, (**b**) any tumor (benign or malignant) vs. control, and (**c**) malignant vs. others (control, benign tumors, and tumor-like lesions). ROMA (purple). Classifier 1 (pink) comprises CA-125, HE4, MRP8/14, and apoptosis parameters. Classifier 2 (blue) incorporates an extended nine-parameter panel, including CA-125, HE4, MRP8/14, apoptosis, OPN, SAA, IL-6, IL-8, and IGFBP-4. The area under the curve (AUC) is shown for each classifier. The multiparametric classifiers display the mean AUC +/− standard deviation from cross-validation, whereas the ROMA AUC is from a direct calculation. The white circles indicate the optimal operating points selected by maximizing Youden’s J index.

**Table 1 diseases-13-00342-t001:** Clinical characteristics of the study participants.

Characteristic	Controls *n* (%)	Benign Tumors *n* (%)	Malignant Tumors *n* (%)	*p*-Value (*)
Ultrasonographic Features				0.0441
Simple	–	11 (12.6%)	0 (0%)	
Complex	–	76 (87.4%)	40 (100%)	
Laterality				0.3322
Unilateral	–	75 (86.2%)	31 (77.5%)	
Bilateral	–	12 (13.8%)	9 (22.5%)	
Tumor Size (mean ± SD)		8.36 ± 4.21	12.18 ± 5.71	<0.0001
<5 cm	–	14 (16.1%)	1 (2.5%)	
5–10 cm	–	54 (62.1%)	18 (45.0%)	
>10 cm	–	19 (21.8%)	21 (52.5%)	

* Values are expressed as *n* (%) for categorical variables and mean ± standard deviation (SD) for continuous variables. The chi-square test was used to compare the proportions of categorical variables between groups (controls, benign tumors, and malignant tumors). One-way ANOVA was used to compare the mean ages between groups. Independent *t*-tests were used to compare tumor sizes between the benign and malignant groups.

**Table 2 diseases-13-00342-t002:** Comparative diagnostic performance of the ROMA and multiparametric classifiers in three clinical scenarios.

Comparison	Classifier	AUC	AUC SD	Optimal Threshold	Sensitivity	Specificity	Youden J
Malignant vs. Benign	ROMA	0.814		51.520	0.771	0.785	0.556
Malignant vs. Benign	Classifier_1	0.831	0.112	0.275	0.800	0.848	0.648
Malignant vs. Benign	Classifier_2	0.880	0.084	0.444	0.800	0.937	0.737
Any Tumor vs. Control	ROMA	0.708		50.050	0.404	1.000	0.404
Any Tumor vs. Control	Classifier_1	0.947	0.036	0.893	0.895	0.889	0.784
Any Tumor vs. Control	Classifier_2	0.917	0.113	0.934	0.833	1.000	0.833
Malignant vs. Others	ROMA	0.821		51.520	0.771	0.807	0.578
Malignant vs. Others	Classifier_1	0.830	0.125	0.270	0.800	0.864	0.664
Malignant vs. Others	Classifier_2	0.896	0.091	0.393	0.829	0.909	0.738

The diagnostic performance of the ROMA was compared to that of two multiparametric classifiers: Classifier 1 (CA-125, HE4, MRP8/14, and Apoptosis) and Classifier 2 (CA-125, HE4, MRP8/14, apoptosis, OPN, SAA, IL-6, IL-8, and IGFBP-4). Performance was evaluated across three clinically relevant scenarios. Metrics for the multiparametric classifiers represent cross-validated results (reported as the mean +/− SD), whereas those for ROMA were obtained by direct calculation.

## Data Availability

The original contributions presented in this study are included in the article. Further inquiries can be directed to the corresponding authors.

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
