# Peer review of "Utility of a Multimodal Biomarker Panel and Serum Proapoptotic Activity to Refine Diagnosis of Ovarian Adnexal Masses"

_diseases, 2025, doi:10.3390/diseases13100342_

Round 1
Reviewer 1 Report
Comments and Suggestions for Authors
Dear Authors,
Thank you for the opportunity to review the manuscript titled: Utility of a multimodal biomarker panel and serum proapoptotic activity to refine the diagnosis of ovarian adnexal masses.
The analysis allowed the authors to demonstrate the usefulness of the biomarkers CA125, MRP8/14, HE4, OPN, SAA, and the ROMA algorithm in reliably differentiating malignant from benign ovarian adnexal masses and controls.
Below are a few minor comments for your consideration.
Part Results:
3.3. Sera from patients with ovarian cancer showed reduced apoptosis-inducing capacity in Jurkat cells
Lines 266-268: ” Since reduced apoptosis in cancer has been linked to elevated soluble CD95 (sCD95/sFAS), we measured sCD95 concentrations; however, no significant differences were detected between groups” – I suggest that the authors comment on this briefly
3.6. Comparative analysis of biomarker divergence across clinical groups
Lines 360-361: ”We also performed a comparative evaluation of the 26 serum biomarkers between control, benign, and malignant ovarian tumor samples” - The Authors included the ROMA test among biomarkers, which, although based on the assessment of the concentration of 2 markers, is an algorithm – I suggest considering a different formulation
Part Disussion:
Lines 472-476: ” Elevated serum sFAS levels have also been observed in patients with breast cancer [43], colon cancer [44], renal [45, 46], cervical, endometrial, and ovarian carcinomas [47]. Our study did not find statistically significant differences in sFAS levels among the control group, benign tumors, and malignant tumors. However, women with tumors tended to have slightly higher levels of this marker”- Are the authors aware of any literature reporting similar levels of sFAS in different types of ovarian tumors?
Author Response
Reviewer 1:
Comments and Suggestions for Authors
Dear Authors,
Thank you for the opportunity to review the manuscript titled: Utility of a multimodal biomarker panel and serum proapoptotic activity to refine the diagnosis of ovarian adnexal masses.
The analysis allowed the authors to demonstrate the usefulness of the biomarkers CA125, MRP8/14, HE4, OPN, SAA, and the ROMA algorithm in reliably differentiating malignant from benign ovarian adnexal masses and controls.
Below are a few minor comments for your consideration.
Part Results:
3.3. Sera from patients with ovarian cancer showed reduced apoptosis-inducing capacity in Jurkat cells
Lines 266-268: ” Since reduced apoptosis in cancer has been linked to elevated soluble CD95 (sCD95/sFAS), we measured sCD95 concentrations; however, no significant differences were detected between groups” – I suggest that the authors comment on this briefly
Response: We are grateful to the reviewer for this valuable observation. As suggested, we have included a brief comment on lines 378 – 380 of the revised manuscript.
3.6. Comparative analysis of biomarker divergence across clinical groups
Lines 360-361: ”We also performed a comparative evaluation of the 26 serum biomarkers between control, benign, and malignant ovarian tumor samples” - The Authors included the ROMA test among biomarkers, which, although based on the assessment of the concentration of 2 markers, is an algorithm – I suggest considering a different formulation
Response: We thank the reviewer for this insightful observation. In response, we have rephrased the relevant section to clarify that our comparative analysis included not only serum biomarkers but also the ROMA algorithm, which is calculated from CA125 and HE4 levels (lines 604-606). We also carefully reviewed the term "ROMA algorithm" throughout the manuscript. This revision prevents any misinterpretation of ROMA as a standalone biomarker while correctly framing it as a diagnostic tool. Furthermore, we have performed additional statistical analyses, the results of which are presented in the new Figure 9 and detailed in the new Table 2.
Part Discussion:
Lines 472-476: ” Elevated serum sFAS levels have also been observed in patients with breast cancer [43], colon cancer [44], renal [45, 46], cervical, endometrial, and ovarian carcinomas [47]. Our study did not find statistically significant differences in sFAS levels among the control group, benign tumors, and malignant tumors. However, women with tumors tended to have slightly higher levels of this marker”- Are the authors aware of any literature reporting similar levels of sFAS in different types of ovarian tumors?
Response: We are grateful to the reviewer for this question. To our knowledge, current literature on sFAS levels across ovarian tumor subtypes is limited and inconsistent. Some studies have reported elevated sFAS in ovarian cancer compared to healthy controls, while others report no significant differences among histological subtypes or between benign and malignant tumors (as mentioned in refs 49 and 50). In response to this comment, we have added a statement in the Discussion section to acknowledge this variability and to emphasize that our findings are in line with reports showing the absence of clear differences in sFAS levels among ovarian tumor types (lines 847-852).

Reviewer 2 Report
Comments and Suggestions for Authors
The authors presented the article entitled “Utility of a multimodal biomarker panel and serum proapoptotic activity to refine diagnosis of ovarian adnexal masses” by analyzing a cohort of 132 participants.
I have suggestions.
Figure 1-6, Tables to show the data, including standard deviation, mean, confidence interval or p value, instead of figures.
Figure 8, choosing the dominant parameters to set a model to predictive ovarian adnexal masses.
Comments on the Quality of English LanguageThe English could be improved to more clearly express the research.
Author Response
Reviewer 2:
The authors presented the article entitled “Utility of a multimodal biomarker panel and serum proapoptotic activity to refine diagnosis of ovarian adnexal masses” by analyzing a cohort of 132 participants.
I have suggestions.
Figure 1-6, Tables to show the data, including standard deviation, mean, confidence interval or p value, instead of figures.
Response: The authors thank the reviewer for this suggestion. While we believe the current figures are essential for the visual interpretation of the results, we acknowledge the importance of providing the underlying data. As such, we have now included the complete dataset (including all molecules, the ROMA index, and the death-induction assay) as Supplementary Table 1.
Figure 8, choosing the dominant parameters to set a model to predictive ovarian adnexal masses.
Response: We are grateful for this comment. As suggested, we performed a new analysis, developing two multiparametric classifiers that integrate the dominant parameters revealed by our prior analyses. Notably, both classifiers significantly surpassed the discriminative capacity of the ROMA algorithm in three clinical scenarios. We have incorporated these results in a new section: 3.8. Development and validation of multiparametric classifiers for enhanced prediction of ovarian adnexal mass (lines 691-770), which include Figure 9 and Table 2. The corresponding methodological details have been added to the Methods section (lines 250-264).
The English could be improved to more clearly express the research.
Response: As suggested, the English language was proofread. We performed further polishing with DeepSeek and then employed the Springer Nature Curie AI-powered scientific writing assistant to revise the entire manuscript (the changes are marked by “Editor 2”).

Reviewer 3 Report
Comments and Suggestions for Authors
Summary
This study investigates the utility of a multimodal biomarker panel for distinguishing benign from malignant ovarian adnexal masses. A cohort of 136 participants was included. Serum levels of established biomarkers (CA125, HE4, ROMA index) and 21 additional circulating molecules were measured. The authors showed that CA125, HE4, ROMA, MRP8/14, OPN, and SAA effectively differentiated malignant from benign tumors. Since CA125, HE4, and ROMA have been studied extensively and their clinical utility is well established, the novelty of this study lies in the evaluation of MRP8/14, OPN, and SAA. However, given that the distribution of histologic subtypes in the malignant cohort differs markedly from established epidemiology, this raises concerns about potential selection bias and the diagnostic performance of the identified biomarkers.
Major concerns-
- In the abstract and methods sections, the authors reported a cohort size of 132; however, in the results section, the cohort size increased to 136. The authors should clarify this discrepancy.
- Introduction: The authors should include a rationale for the selection of the 21 molecules evaluated in this study. Currently, it is unclear why these molecules were included in addition to established biomarkers. Please Include a brief explanation, such as biological relevance, previous evidence, hypothesis-driven, or clinical applicability, which will help justify their inclusion and clarify the scientific basis for the multimodal biomarker panel.
- Section 3.1: The authors reported “In contrast, the malignant group showed an opposite trend: 35.0% were premenopausal women (mean age 42.6 years), compared to 65.0% postmenopausal women (mean age 56.3 years).” However, in figure 1, the premenopausal women is 37.5% and the postmenopausal women is 62.5%. Please clarify the discrepancy.
- According to real world registry data, among patient with malignant ovarian adnexal masses, HGSOC is usually ~70% of all ovarian carcinomas, Endometrioid: ~10–12%; Mucinous: ~3–11%; Clear cell: ~5–10%; Low-grade serous: ~5%; Sex cord–stromal: ~2–6%. However, in this study, the distribution of histologic subtypes in the malignant cohort differs markedly from established epidemiology, with a lower proportion of high-grade serous carcinoma and higher proportions of mucinous, clear cell, and endometrioid carcinomas. This raises concerns about selection bias because diagnostic performance of biomarkers can vary significantly by histologic subtype. The authors should discuss how this distribution may affect the generalizability of their findings.
- Section 3.3: The experiment of apoptosis-inducing capacity of patient sera in Jurkat cells is somewhat disconnected from the rest of the results. This part seems dilute the focus of the paper and is hard to follow in the current context. The rationale for including this experiment is not clearly integrated with the study’s main objective of evaluating circulating biomarkers for diagnostic purposes. The text references prior work (“we have previously reported…”) and provides descriptive statistics without clearly linking the findings to the multimodal biomarker panel or clinical outcomes. Additionally, the authors report a reduction of apoptosis in malignant sera, but no significant differences in sCD95 levels, which further raises questions about the mechanistic interpretation. The authors should clarify how these findings complement the biomarker panel, whether apoptosis provides independent diagnostic information, and focus on novel insights rather than previously published results.
- Section 3.5: There are several issues in the way the authors presented and written this section. For statistical reports, sometimes they report means and sometimes medians. For example, IL-6 and IL-8 are reported as means, while MRP8/14 is reported as a median. This is problematic because non-parametric tests (KW, Mann–Whitney) is used, which should be based on medians/ranks. Reporting means in that context is misleading. In addition, the formatting of the values is inconsistent. There are at least three different styles used to present mean/median (ex. “mean 4.54 pg/mL”, “mean: 1.98 pg/mL”, and “28.80 pg/mL”). The author should use one consistent approach/format (preferably medians + IQR or range) since that aligns with their statistical tests.
- Section 3.5: The authors mentioned that 21 molecules were evaluated, but only 10 are presented in figure 5&6, which leaves the reader confused. The authors should either present all 21 molecules (in supplementary figures) or explain why other 11 biomarkers were excluded.
- Section 3.7: The authors included apoptosis in the analysis; however, apoptosis in the section 3.3 is reported as the percentage, unlike other biomarkers presented in absolute concentrations. The authors should clarify how this functional readout was scaled or standardized for inclusion in comparative analyses (ex. ROC curves, effect sizes). Additional explanation on the rationale for treating apoptosis as a quantitative biomarker and its comparability with molecular measurements should also be provided.
- Since no validation experiments were performed, I recommend that the authors provide guidance on the potential clinical utility of their findings in the discussion section. For example, suggest how to better distinguish benign from malignant ovarian masses in practice, which combination of biomarkers or functional assays might be most effective, and how their multimodal panel could be integrated with existing diagnostic tools such as CA125, HE4, ROMA, and imaging. Including such discussion would help readers understand the translational relevance and practical implementation of the study.
Minor concerns -
- Ref [3] Siegel RL, Miller KD, Jemal A. Cancer statistics, 2019. CA Cancer J Clin. 2019;69(1):7-34 and ref [27] Siegel RL, Miller KD, Fuchs HE, Jemal A. Cancer statistics, 2022. CA Cancer J Clin. 2022;72(1):7-33 appear to serve the same purpose. The authors should cite the most recent Cancer Statistics, 2025, instead.
- Line 202: “Boxsplot” should be correct to “Boxplot”.
- Line 226: “(p-value = 0.0000)” is confusing. Please fix.
- Line 227: “(12.18 ± 5.715.71 cm)” is confusing. Please fix it.
Author Response
Reviewer 3:
This study investigates the utility of a multimodal biomarker panel for distinguishing benign from malignant ovarian adnexal masses. A cohort of 136 participants was included. Serum levels of established biomarkers (CA125, HE4, ROMA index) and 21 additional circulating molecules were measured. The authors showed that CA125, HE4, ROMA, MRP8/14, OPN, and SAA effectively differentiated malignant from benign tumors. Since CA125, HE4, and ROMA have been studied extensively and their clinical utility is well established, the novelty of this study lies in the evaluation of MRP8/14, OPN, and SAA. However, given that the distribution of histologic subtypes in the malignant cohort differs markedly from established epidemiology, this raises concerns about potential selection bias and the diagnostic performance of the identified biomarkers.
Major concerns-
- In the abstract and methods sections, the authors reported a cohort size of 132; however, in the results section, the cohort size increased to 136. The authors should clarify this discrepancy.
Response: We thank the reviewer for identifying this discrepancy. The cohort size has been thoroughly reviewed and updated to consistently reflect a total of 136 participants (9 controls, 87 benign adnexal masses, and 40 malignant adnexal masses). These corrections have been incorporated throughout the manuscript, including the Abstract, Methods, and Results sections.
- Introduction: The authors should include a rationale for the selection of the 21 molecules evaluated in this study. Currently, it is unclear why these molecules were included in addition to established biomarkers. Please Include a brief explanation, such as biological relevance, previous evidence, hypothesis-driven, or clinical applicability, which will help justify their inclusion and clarify the scientific basis for the multimodal biomarker panel.
Response: We thank the reviewer for this comment. In response, we have expanded the Introduction (lines 110-128) to provide a more detailed description of the measured parameters. The revised text now specifies that the analysis included 24 soluble molecules, the ROMA algorithm (derived from CA125 and HE4 levels), and an independent apoptosis assay. It also clarifies that these molecules were selected based on their established biological relevance, prior evidence in ovarian cancer and other malignancies, and their potential clinical applicability as diagnostic or prognostic biomarkers.
- Section 3.1: The authors reported “In contrast, the malignant group showed an opposite trend: 35.0% were premenopausal women (mean age 42.6 years), compared to 65.0% postmenopausal women (mean age 56.3 years).” However, in figure 1, the premenopausal women is 37.5% and the postmenopausal women is 62.5%. Please clarify the discrepancy.
Response: We thank the reviewer for pointing out this discrepancy and apologize for the oversight. The text has been updated to align with Figure 1, now accurately reporting 37.5% premenopausal women (mean age 43.9 years) and 62.5% postmenopausal women (mean age 56.1 years).
- According to real world registry data, among patient with malignant ovarian adnexal masses, HGSOC is usually ~70% of all ovarian carcinomas, Endometrioid: ~10–12%; Mucinous: ~3–11%; Clear cell: ~5–10%; Low-grade serous: ~5%; Sex cord–stromal: ~2–6%. However, in this study, the distribution of histologic subtypes in the malignant cohort differs markedly from established epidemiology, with a lower proportion of high-grade serous carcinoma and higher proportions of mucinous, clear cell, and endometrioid carcinomas. This raises concerns about selection bias because diagnostic performance of biomarkers can vary significantly by histologic subtype. The authors should discuss how this distribution may affect the generalizability of their findings.
Response: We thank the reviewer for highlighting this important point. The epidemiological data from our study originate from a specific tertiary-care hospital within the Mexican Institute of Social Security (IMSS) in Western Mexico, reflecting the regional patient population. We agree that the histological subtype distribution in our cohort differs from the global average. However, this distribution is consistent with well-documented geographic variations. Our data align with reported profiles from countries like Turkey (ref 39) and Italy (ref 41). Precisely for this reason, our data are contrasted with findings from other regions in the Discussion section (lines 796 to 804). In response to the reviewer’s concerns, we have now added a sentence to the Discussion (lines 804-807) to contextualize our findings within this framework of geographic variation. This addition clarifies that our results may be particularly generalizable to populations with similar histologic profiles.
- Section 3.3: The experiment of apoptosis-inducing capacity of patient sera in Jurkat cells is somewhat disconnected from the rest of the results. This part seems dilute the focus of the paper and is hard to follow in the current context. The rationale for including this experiment is not clearly integrated with the study’s main objective of evaluating circulating biomarkers for diagnostic purposes. The text references prior work (“we have previously reported…”) and provides descriptive statistics without clearly linking the findings to the multimodal biomarker panel or clinical outcomes. Additionally, the authors report a reduction of apoptosis in malignant sera, but no significant differences in sCD95 levels, which further raises questions about the mechanistic interpretation. The authors should clarify how these findings complement the biomarker panel, whether apoptosis provides independent diagnostic information, and focus on novel insights rather than previously published results.
Response: We thank the reviewer for this comment and the opportunity to clarify. To address this point, we have expanded the introduction (lines 110-128) to better integrate the apoptosis measurements with the evaluation of circulating molecules. A corresponding paragraph has been added to the discussion (lines 1164-1184) to link these concepts. The apoptosis assay provides functional insight into the serum's proapoptotic or proinflammatory state, complementing the quantitative data from the circulating biomarkers. Specifically, the observed reduction of apoptosis in malignant sera suggests alterations in serum-mediated immune signaling that may not be apparent when measuring individual biomarkers alone.
- Section 3.5: There are several issues in the way the authors presented and written this section. For statistical reports, sometimes they report means and sometimes medians. For example, IL-6 and IL-8 are reported as means, while MRP8/14 is reported as a median. This is problematic because non-parametric tests (KW, Mann–Whitney) is used, which should be based on medians/ranks. Reporting means in that context is misleading. In addition, the formatting of the values is inconsistent. There are at least three different styles used to present mean/median (ex. “mean 4.54 pg/mL”, “mean: 1.98 pg/mL”, and “28.80 pg/mL”). The author should use one consistent approach/format (preferably medians + IQR or range) since that aligns with their statistical tests.
Response: We apologize for these oversights. Section 3.5 has been revised to report all values consistently as means, and the formatting has been standardized throughout the section to ensure clarity and uniformity. Furthermore, the complete dataset, including all molecules, the ROMA index, and the death-induction assay, has been included as Supplementary Table 1.
Section 3.5: The authors mentioned that 21 molecules were evaluated, but only 10 are presented in figure 5&6, which leaves the reader confused. The authors should either present all 21 molecules (in supplementary figures) or explain why other 11 biomarkers were excluded.
Response: We thank the reviewer for this comment. The revised manuscript now specifies that the analysis included 24 soluble molecules, the ROMA algorithm (derived from CA125 and HE4 levels), and an independent apoptosis assay.
As suggested, we have clarified that the remaining biomarkers were excluded from the figures as they showed no significant differences between the study groups (lines 484-486). Furthermore, complete statistical data for all molecules, including the 14 that showed no significant variation, are presented in Supplementary Table 1.
- Section 3.7: The authors included apoptosis in the analysis; however, apoptosis in the section 3.3 is reported as the percentage, unlike other biomarkers presented in absolute concentrations. The authors should clarify how this functional readout was scaled or standardized for inclusion in comparative analyses (ex. ROC curves, effect sizes). Additional explanation on the rationale for treating apoptosis as a quantitative biomarker and its comparability with molecular measurements should also be provided.
Response: We thank the reviewer for this important comment and the opportunity to clarify this point Each parameter, including the apoptosis measurement, was analyzed individually on its native scale for both the divergence analysis (Kruskal-Wallis test) and the ROC curve analysis. The non-parametric Kruskal-Wallis test was employed to assess differences between the three groups, this test is suitable for comparing groups of data that are on different scales. The effect size (eta-squared, η²) for this test was calculated using the eta-h-squared formula derived from Kruskal-Wallis H-statistic value, this metric indicates the proportion of total variance explained by this independent variable, this assessment is independent to the directionality of the parameter´s correlation or scale. For the diagnostic performance evaluation of individual parameters in ovarian adnexal masses identification, individual ROC curves were employed for each measurement; in the case of apoptosis, classes were inverted to align with its negative correlation to ovarian adnexal massed and malignancy. The 2.7. statistical analysis section was complemented (lines 250 - 264).
- Since no validation experiments were performed, I recommend that the authors provide guidance on the potential clinical utility of their findings in the discussion section. For example, suggest how to better distinguish benign from malignant ovarian masses in practice, which combination of biomarkers or functional assays might be most effective, and how their multimodal panel could be integrated with existing diagnostic tools such as CA125, HE4, ROMA, and imaging. Including such discussion would help readers understand the translational relevance and practical implementation of the study.
Response: We thank the reviewer for this valuable suggestion. In response to this and similar feedback from another reviewer, we performed a new analysis to develop two multiparametric classifiers based on the dominant parameters identified in our study.
Notably, both classifiers significantly outperformed the ROMA algorithm across three clinical scenarios. These new findings are presented in a dedicated section, 3.8. Development and validation of multiparametric classifiers for enhanced prediction of ovarian adnexal masses (lines 691-770), which includes the new Figure 9 and Table 2. The corresponding methodological details have been added to the Methods section (lines 250-264), and we have expanded the Discussion to cover the potential clinical applications of these classifiers (lines 1164-1184).
Minor concerns -
- Ref [3] Siegel RL, Miller KD, Jemal A. Cancer statistics, 2019. CA Cancer J Clin. 2019;69(1):7-34 and ref [27] Siegel RL, Miller KD, Fuchs HE, Jemal A. Cancer statistics, 2022. CA Cancer J Clin. 2022;72(1):7-33 appear to serve the same purpose. The authors should cite the most recent Cancer Statistics, 2025, instead.
Response: We thank the reviewer for pointing this out. We have updated the manuscript to cite the most recent cancer statistics (Siegel RL et al., CA Cancer J Clin, 2025) instead of the previous 2019 and 2022 references. The Introduction and relevant sections now reflect the most current epidemiological data.
- Line 202: “Boxsplot” should be correct to “Boxplot”.
Response: We thank the reviewer for pointing out this typographical error. The term “Boxsplot” has been corrected to “Boxplot” in line 276 of the new manuscript version.
- Line 226: “(p-value = 0.0000)” is confusing. Please fix.
Response: We thank the reviewer for pointing this out. The p-value previously reported as “0.0000” has been corrected to “<0.0001” in Table 1 to accurately reflect statistical reporting.
- Line 227: “(12.18 ± 5.715.71 cm)” is confusing. Please fix it.
Response: We appreciate the reviewer’s careful observation. The error has been corrected, and the value now reads “12.18 ± 5.71 cm.” (line 325 of the new manuscript version).

Round 2
Reviewer 3 Report
Comments and Suggestions for Authors
I appreciate the authors' thorough responses to the concerns raised in my review. They have addressed all of the issues, and the manuscript is significantly improved as a result. I have no further concerns and support its publication.